# An epigenetic change in a moth is generated by temperature and transmitted to many subsequent generations mediated by RNA

**Jaroslav Pavelka**[1] *, **Simona Poláková**[2], **Věra Pavelková**[3], **Patrik Galeta**[4]

**1** University of West Bohemia, Centre of Biology, Pilsen, Czech Republic, **2** Ministry of the Environment of the Czech Republic, Praha, Czech Republic, **3** Faculty of Science, University of South Bohemia, České Budějovice, Czech Republic, **4** Department of Anthropology, University of West Bohemia, Pilsen, Czech Republic

* Japetos@seznam.cz

**Data Availability Statement:** All relevant data are within the paper and its Supporting information files.

## Abstract

Epigenetic changes in sexually reproducing animals may be transmitted usually only through a few generations. Here we discovered a case where epigenetic change lasts 40 generations. This epigenetic phenomenon occurs in the short antennae (*sa*) mutation of the flour moth (*Ephestia kuehniella*). We demonstrate that is probably determined by a small RNA (e.g., piRNA, miRNA, tsRNA) and transmitted in this way to subsequent generations through the male and female gametes. The observed epigenetic change cancels *sa* mutation and creates a wild phenotype (a moth that appears to have no mutation). It persists for many generations (40 recorded). This epigenetic transgenerational effect (suppression homozygous mutation for short antennae) in the flour moth is induced by changes during ontogenetic development, such as increased temperature on pupae development, food, different salts in food, or injection of RNA from the sperm of already affected individuals into the eggs. The epigenetic effect may occasionally disappear in some individuals and/or progeny of a pair in the generation chain in which the effect transfers. We consider that the survival of RNA over many generations has adaptive consequences. It is mainly a response to environmental change that is transmitted to offspring via RNA. In this study, we test an interesting epigenetic effect with an unexpected length after 40 generations and test what is its cause. Such transfer of RNA to subsequent generations may have a greater evolutionary significance than previously thought. Based on some analogies, we also discuss of the connection with the SIR2 gene.

## Introduction

The epigenetic inheritance seems to depend on the inheritance of acquired phenotypic alterations without any change in the sequence of the DNA, while the epigenetic changes may frequently be erased after few generations. However, in some cases, epigenetic information can be transmitted for several generations from parent to progeny (multigenerational epigenetic

**Funding:** The author(s) received no specific funding for this work.

**Competing interests:** The authors have declared that no competing interests exist.

inheritance) [1]. The observation that environmental stress can also promote transgenerational pathologies suggests ancestral stress conditions may be a significant factor in our own disease and what we pass down to our descendants [2]. Exposure to the environmental stressors can induce various epigenetic changes (epimodifications) in mammalian germ cells, which can influence the developmental trajectory of the offspring across generations [3]. The acquired characteristics can occur through ancestral exposures or experiences and certain paternally acquired traits can be 'memorized' in the sperm as epigenetic changes triggered by epigenetic molecular mechanisms and post-translational modifications [2].

Transgenerational inheritance works in such a way that only one generation is affected by some factors and the effect is then manifested in other generations that are not affected by the factor in any way. In the case of transgenerational epigenetic inheritance, none of the genetic material of the descendants was present or exposed to the initiating environmental or genetic signal [4].

We can distinguish pre- and post-transcriptional epigenetic mechanisms. Methylation of DNA and modification of histones regulate transcription (pre-transcriptional), and mechanisms such as ubiquitinization, autophagy and microRNAs regulate development post-transcriptionally [5]. Nevertheless, several possibilities can be classified as post-transcriptional epigenetic mechanisms: microRNA, tRNA, piRNA, possible methylation, or some other effects. For instance, Gapp et al. [6] described that the traumatic stress in early life modified mouse microRNA (miRNA) expression and behavioral and metabolic responses in the progeny. Therefore, stress can induce the transgenerational inheritance of disease, and ancestral exposures to a variety of factors can alter stress response transgenerationally. In *Drosophila melanogaster*, the phenotypic defects of wings caused by cadmium can be inherited to the offspring, and this transgenerational inheritance effect may be related to the epigenetic regulation of histone methylation. Therefore, the adaptability of offspring should be considered when evaluating the toxicity and environmental risk of cadmium [7]. In a similar study [8] of *D. melanogaster*, cadmium altered larval body length and weight, increased pupation and eclosion times, and altered the relative expression levels of development-related genes [8]. The results showed that the delayed effects of pupation and eclosion time could be maintained for two generations, and the inhibitory effects of juvenile hormone (JH) and ecdysone (20-hydroxyecdysone, 20E) could be maintained for two or three generations [8]. Cadmium increased the expression of genes related to DNA methylation (dDnmt2, dMBD2/3) in ovary (F0–F2) and testis (F0 and F1) [8].

Insecticide-induced effects can be transgenerationally inherited. These are heritable epigenetic modifications that respond to pesticide and xenobiotic stress. Therefore, pesticides can control the development of resistance through epigenetic processes. Additionally, pesticide-activated insect pests can better tolerate additional stress, further increasing their success in adapting to agroecosystems [9]. Transgenerational effects are common in species living in habitats subjected to recurrent stressful events, such as fluctuating resource availability. The nutritional status of the midge *Chironomus tepperi* has been reported to influence life history traits of the offspring. Offspring of parents reared under low food conditions had a shorter development time and lower reproductive output compared to offspring of parents raised under excess food [10].

In addition to the above-mentioned causes of transgenerational epigenetic inheritance, there are also examples of histone modification, DNA methylation, the influence of sRNA, or a combination of various factors [4]. Histone modifications can be transmitted through cell division and generations by multiple methods including non-coding RNA [4]. Gene regulation is maintained by epigenetic mechanisms including DNA methylation, histone modifications and non-coding RNAs. These same mechanisms are responsible for silencing of transposable

elements and heterochromatin formation [11]. Interestingly, epigenetic mechanisms can transmit the transcriptional state of a gene to the next generation [11]. The role for DNA-bound proteins in epigenetic inheritance has been extensively demonstrated. Sperm histones (like somatic histones) carry post-translational modifications. For example, the trithorax mark histone H3 lysine 4 trimethylation (H3K4me3) and the polycomb mark histone H3 lysine 27 trimethylation (H3K27me3) are important for the normal development and the link with the maintenance of transcription patterns [12].

The tRNA-derived RNA fragments (tRFs, mse-tsRNA) are sometimes responsible for the epigenetic inheritance. The tRFs are the most abundant class of RNAs in mature sperm [13, 14]. In addition RNA methyltransferase (Dnmt2) generates methylation of the tRNA. It has been associated with the epigenetic phenomenon for father to offspring transmission [15]. The traumatic stress in early life can cause upregulation of miRNAs in F1, affection of piRNA in particular cluster in sperm. Additonally, piRNA cluster causes the complex changes in stress-coping behaviors, metabolism and stress-induced glucose release in the offspring [6, 14]. In *D. melanogaster* it has been shown that long-term adaptation may affect miRNA profiles in sperm and that these may show varied interactions with short-term environmental changes [16]. Organisms appear to protect certain RNAs by design. Unlike target-directed degradation of microRNAs, complementarity-dependent destabilization of piRNAs in flies, 2'-O-methylation also protects small interfering RNAs (siRNAs) from complementarity-dependent destruction [17]. RNA harbored by mammalian sperm is increasingly considered to be an additional source of paternal hereditary information, beyond DNA. Recent studies have demonstrated the role of sperm small noncoding RNAs (sncRNAs) in modulating early embryonic development and offspring phenotype [18]. Another epigenetic mechanism is the addition of poly (UG) ("pUG") repeats to the 3' ends of mRNAs, where they drives gene silencing and transgenerational epigenetic inheritance in the metazoan *Caenorhabditis elegans*. The pUG tails promote silencing by recruiting an RNA-dependent RNA Polymerase (RdRP) that synthesizes small interfering (si)RNAs [19]. Many sRNAs are unusual in that they can be produced in two ways, either using genomic DNA as the template (primary sRNAs) or existing sRNAs as the template (secondary sRNAs). Thus, organisms can evolve rapid plastic responses to their current environment by adjusting the amplification rate of sRNA templates. The sRNA levels can also be transmitted transgenerational by the direct transfer of either sRNAs or the proteins involved in amplification [20].

Although small and transcriptionally inert, sperm cell with extremely compacted genome and virtually no cytoplasm contains a plethora of small RNAs and a large number of DNA sequences packaged by histones and a distinctive DNA methylation profile [12]. Nevertheless, dysregulation of at least two different microRNAs (miR-1 and miR-124) in sperm and their transmission to the egg have been postulated to be the causes of two cases of intergenerational inheritance in mouse [21, 22]. Transgenerational epigenetic inheritance has been described for several lineages. For instance Forneck et al. [23] found over 100 cases of epigenetic inheritance in 42 different species (bacteria, protists, plants, animals). However, the number of generations with epigenetic characteristics (epigenetic memory) are usually restricted. Rassoulzadegan et al. [24] described that mouse effect of white spots, which is caused by miRNA, disappears after six generations. In grape phylloxera *Daktulosphaira vitifoliae* Fitch, 1855 (Hemiptera: Phylloxeridae) epigenetic memory was observed for 15 generations [17], however, in this case it was parthenogenetically inherited. Apparently, small RNAs apply in many of these cases. It appears that small RNAs do not function without assistance. Buckley et al. [1] described that the defective HRDE-1, encoded by Argonaute protein, directs gene-silencing events in germ-cell nuclei that drive multigenerational RNAi inheritance and promote long-term resistance of the germ-cell lineage.

In our previous study [25], we found transgenerational epigenetic inheritance in the Mediterranean flour moth [*Ephestia kuehniella* Zeller (Lepidoptera: Pyralidae)], but we did not find the molecular mechanisms behind this process, which can be caused by a number of causes (see below). A specific phenomenon of phenotypic inheritance (paramutation, epigenetic heredity) was assessed for the first time in the short antennae *(sa)* morphological mutation of *E. kuehniella*. The *sa* mutation is inherited as a simple autosomal recessive gene and causes considerable shortening of antennae in moths (Fig 1A) of both sexes [25]. At higher temperature, short antennae of *sa* moths revert to a normal non-mutant (wild) phenotype with long antennae (Fig 1B) and it rarely returns to the original state (*sa*). Although a genotype remains the same, the change is transmitted to subsequent generations [25] and appears in subsequent generations even at low temperature. The epigenetic effect suppresses the effect of the *sa* mutation and the *sa* mutant is indistinguishable from the wild type, therefore we named it *sa*$^{\text{WT}}$ (*sa* wild type) (Fig 1B). Individuals with shortened antennae are at a disadvantage when searching for a sexual partner. They probably have a broken sense of smell, the receptors of which are on the antennae [25]. This mutation apparently reduces fitness.

In this study, we focus on the cause of an epigenetic phenomenon in *Ephestia kuehniella* butterfly (reversion from mutant short *sa* antennae to wild type long *sa*$^{\text{WT}}$ antennae) and duration of its persistence over generations. First, we try to identify molecular mechanisms of the inheritance of this epigenetic phenomenon. We dissected the sperm sac that the male gives to the female, divide it into individual components and inject them in homogenized form into already fertilized eggs from a population with the *sa* mutation. Based on the experiment, we demonstrate transmission to the future generations via RNA. Second, we try to identify other environmental causes of epigenetic change. In addition to the already tested higher

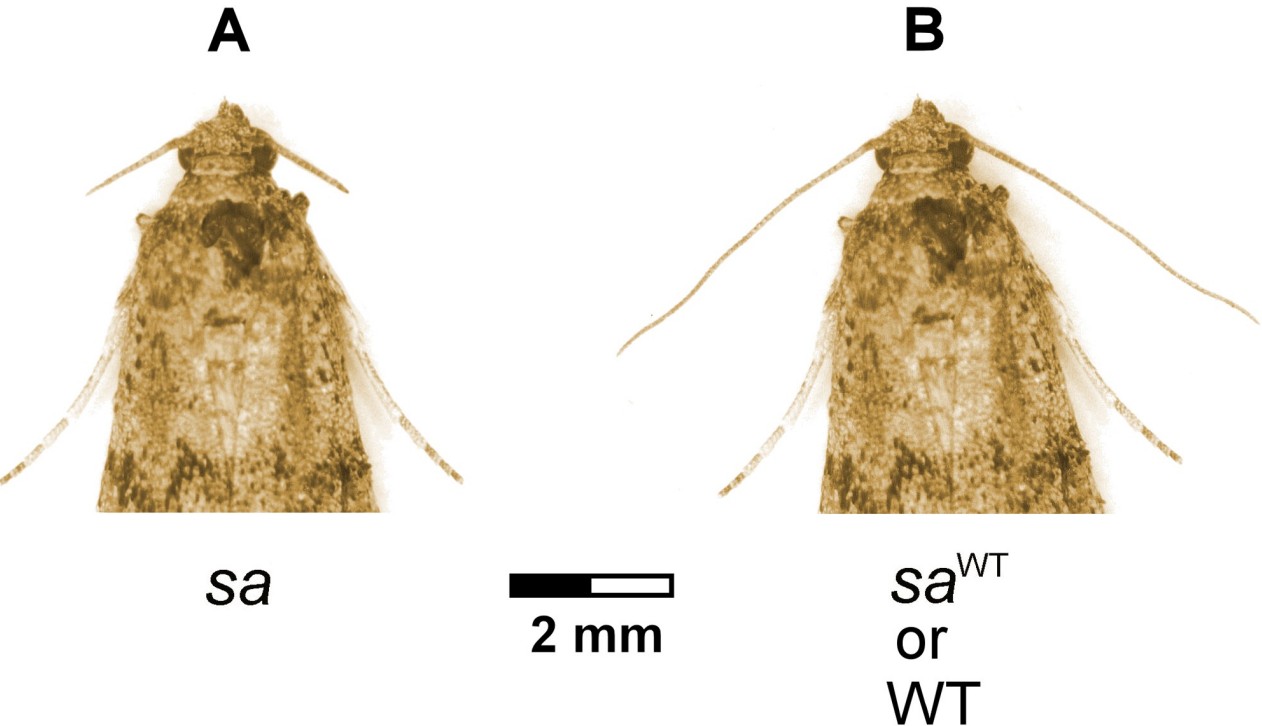

**Fig 1. Phenotypic categories of sa mutation in *Ephestia kuehniella* evaluated according to the expresion of the *sa* allele.** (A) *sa* moth with normal *sa* phenotype (the *sa* allele standard expressed). (B) (*sa*$^{\text{WT}}$ or normal wild type) moth with long antennae undistinguishable from those of wild-type moth (the *sa* phenotype entirely suppressed).

temperature, we examine the effect of feeding stress-inducing chemicals and less nourishing food. Third, we study the duration and stability of the epigenetic phenomenon over many generations and the frequency of the return to the original *sa* phenotype.

## Materials and methods

### Flour moth rearing and handling

**Animals and breeding.** For experiments, the strain of the Mediterranean flour moth *(Ephestia kuehniella* Zeller, Lepidoptera: Pyralidae) homozygous for the autosomal recessive mutation *short antennae* (*sa*) was used. The short (*sa*) and long antennae (wild type, *sa*$^{WT}$) can be seen in Fig 1. Wild type and epigenetically altered mutant antennae (*sa*$^{WT}$) are morphologically indistinguishable. The strain was derived from a mutant Qy strain that was obtained from the stock cultures of W. B. Cotter, Jr. (Albert B. Chandler Medical Center, University of Kentucky, Lexington, KY) and has been kept in single-pair cultures at the Institute of Entomology (České Budějovice, Czech Republic) since 1991.

Stock cultures were reared in constant temperature rooms (20˚C ± 1˚C) at a 12-h:12-h light/dark regime, and at a humidity level of about 40%. Experimental and control cultures were kept at either 20˚C ± 1˚C at the same conditions. New generations were reared from single-pair cultures. Pairs were collected during copulation and placed individually in empty Petri dishes (6 cm in diameter). Females laid eggs for 3–4 days, then imagoes were removed and Petri dishes with eggs were put into plastic boxes with food. Hatching larvae migrated from the dish to the food where they completed their development. Larvae were fed with milled wheat grains supplemented with a small amount of dried yeast. Insects were killed in a container with 96% ethanol.

**Procedure.** In order to determine the epigenetic effect we utilized multiple flour moths with an epigenetic effect, kept in normal breeding conditions. We observed the ratio of short and long antennae in every generation. The exact number of *E. kuehniella* with a specific phenotype was observed until F12. The ratio of phenotypes was relatively constant, and subsequent generations (F13–F40) were simply observed visually without a complete count being made.

Unlike the previous study [25], we distinguished only two categories: short antennae (*sa*) and long antennae (*sa*$^{WT}$). In the first generation, an epigenetic phenotype (*sa*$^{WT}$) was created by treating moths at 25˚C ± 1˚C, which phenotypically silenced the sa mutation. Subsequent generations with the wild-type phenotype (*sa*$^{WT}$) were then nursed at 20˚C to exclude additional influence of temperature, and to monitor epigenetic feature only. The *sa* males of the wild phenotype (*sa*$^{WT}$) were used for all subsequent generations. Each male of a *sa*$^{WT}$ phenotype (*sa* genotype) was mated to a randomly chosen virgin female of the same phenotype and genotype.

To find out what is involved in the epigenetic modification, the male spermatophore (from *sa*$^{WT}$ male) was dissected and divided into individual parts (sperm, product of accessory gland, homogenized fractions of sperm, homogenized spermatophore sac, homogenized sperm with RNase, and total RNA), and these individual parts were then injected into pre-fertilized eggs. The eggs were from the original *sa* line, male and female *sa* phenotype, i.e., the line with a normal manifestation of short antennae mutation.

The experimental design was governed by the method of exclusion: at first the impact of separate ejaculate components of *sa*$^{WT}$ males (spermatophore envelope, homogenized spermathopore sac, product of accessory gland) was tested. Initially, we focused on components without sperm (homogenized spermatophore sac and product of accessory gland). Then, we used wholle sperm, homogenized fraction of sperm (both with and without RNAase), and

finally total RNA isolated from the sperm of $sa^{WT}$ males was used. Finally, eggs were injected with a solution of geldanamycin and, for control, with a clean buffer (a buffer that was also used for injections—in the other experiments, homogenized spermatozoa, RNA, etc. were dissolved in it).

**Isolation of separate parts of spermatophore.** Immediately after copulation ended ($sa^{WT}$ males and females), females were dissected. We knew from previous experiments [25] that the as-yet-unknown substance that causes the epigenetic effect is transmitted more significantly along the male line than along the female line, and easier extraction was offered from the spermatophore. Therefore, $sa^{WT}$ males were used. The spermatophore of males was removed from the females after copulation. The spermatophore was divided into three parts using sharp microtweezers (Fig 2). The following components from male spermatophore were separated for injection: (1) the spermatophore sac from the bursa copulatrix, (2) the seminal fluid from the spermatophore containing both the eupyrene and apyrene sperm (Fig 3), and (3) the secretion of male accessory glands from the bursa copulatrix (Fig 2). Ten samples of each category were stored at −70˚C, and later homogenized and mixed with injection buffer

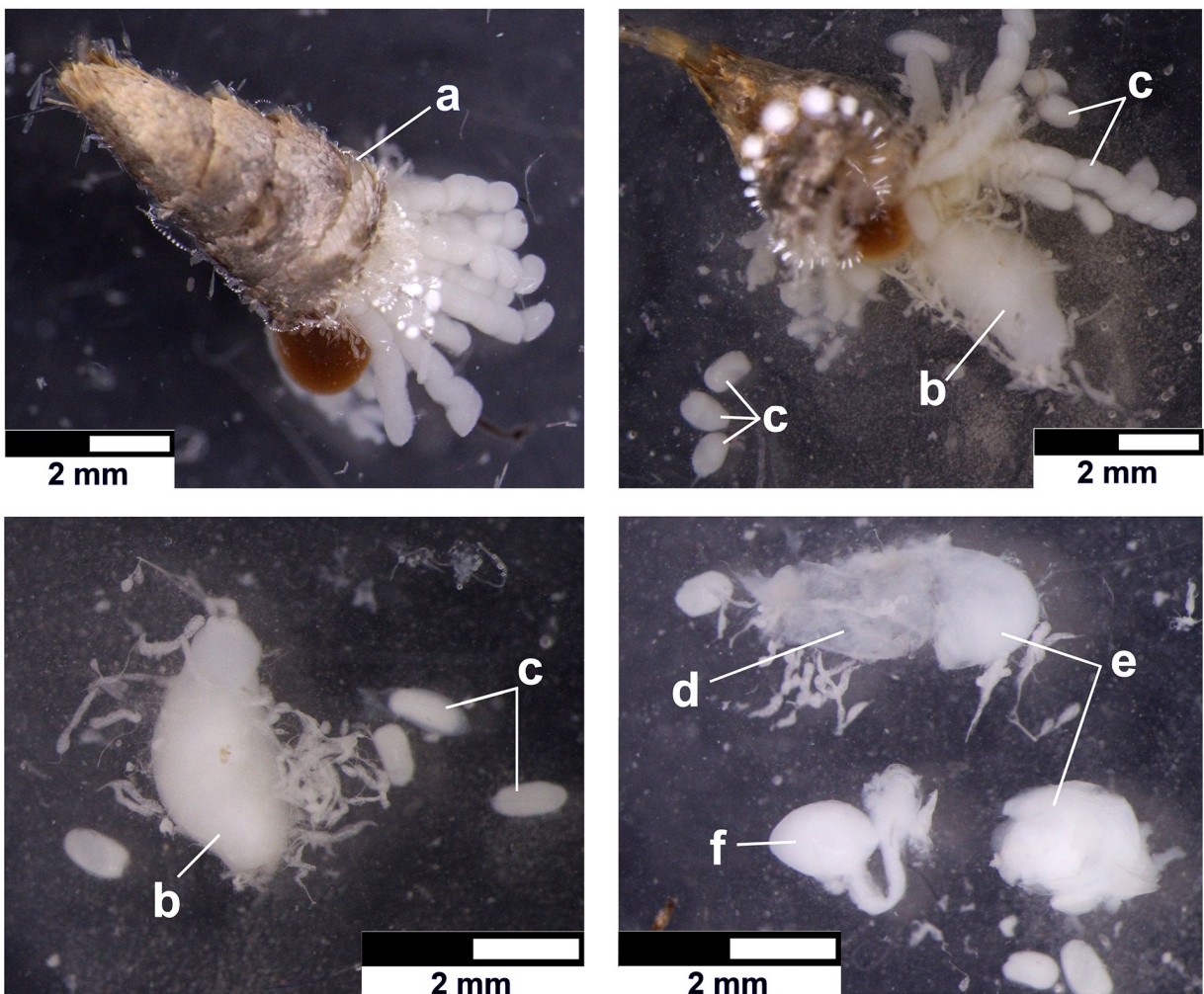

**Fig 2. The dissected abdomen of a just-fertilized *E. kuehniella* female.** (a) abdomen; (b) sperm sac; (c) unfertilized eggs; (d) dissected empty sperm sac; (e) accessory gland product. (f) spermatophore with sperm.

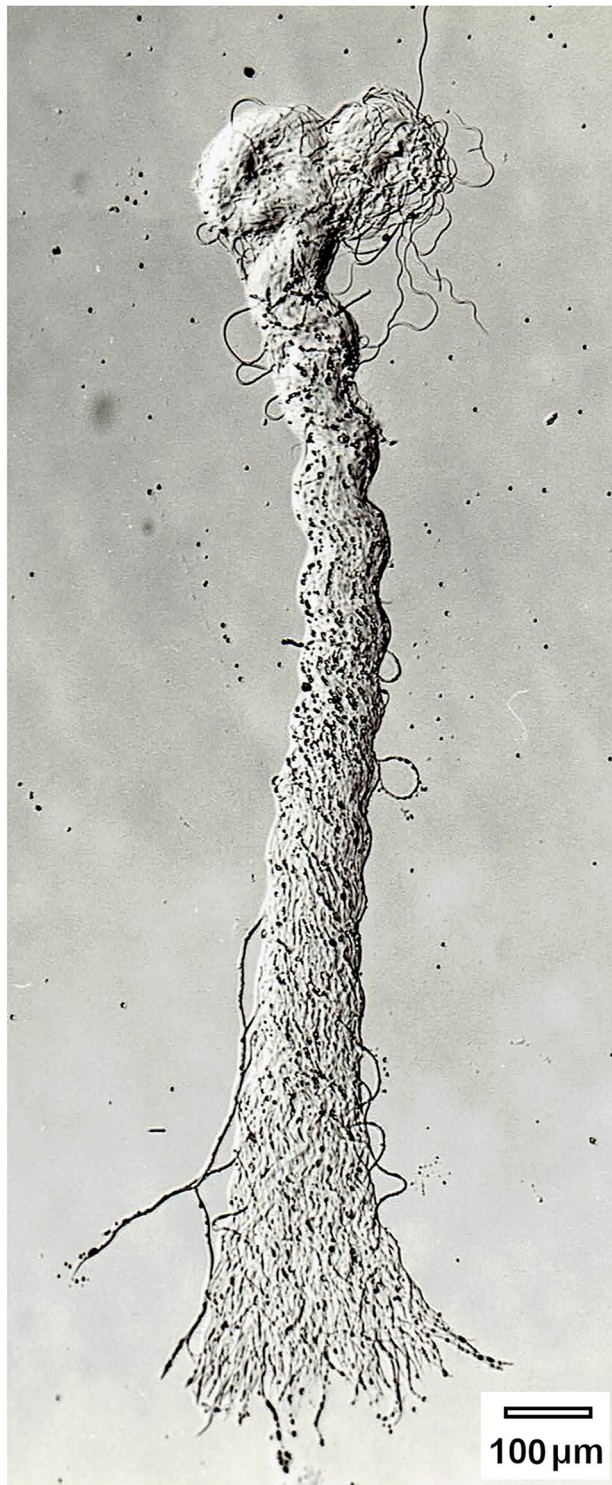

**Fig 3. A bundle of eupyrene sperms of *E. kuehniella* contains an average of 250 spermatozoa- modified according to Koudelová and Cook [61].**

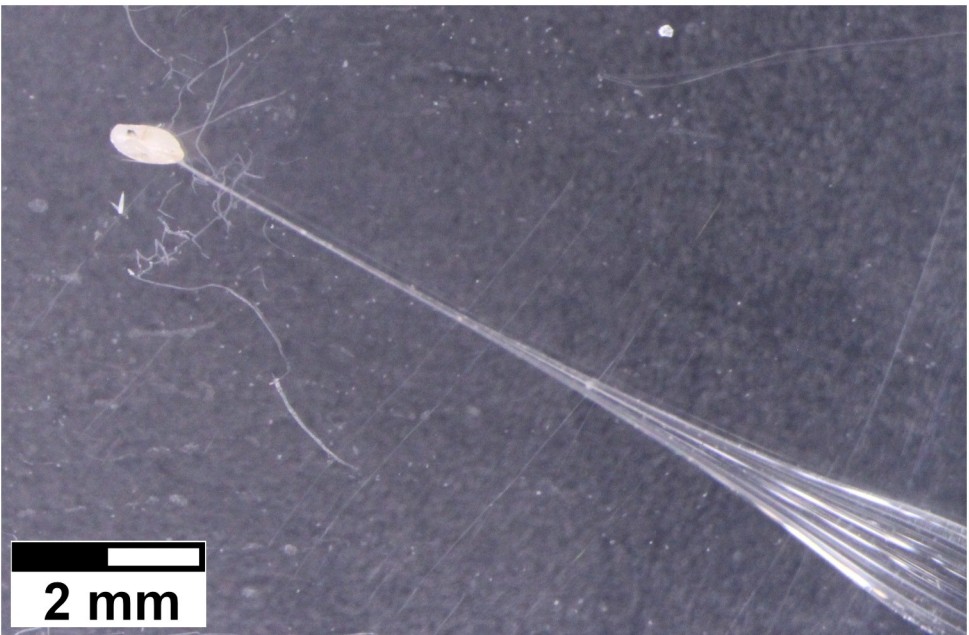

**Fig 4. Injection into the egg of *E. kuehniella*.**

(5 mM KCl, 0.1 mM NaH$_2$PO$_4$, pH 6.8) [26]. Separated parts of spermatophore were prepared in room temperature. Only sperm into which RNase was added after homogenization (inactivated after 3 hours by incubating at 95˚C for 30 seconds) were used for injections. Sperm proteins were purificated using Sep-Pak® (Cartridges for solid Phase Extraction) Waters Corporation according to the manufacturer's instructions. The acquired components were mixed together with injection buffer [26] and individual dissolutions were divided into several aliquots and deep frozen at –70˚C. These aliquots have been used successively to inject fertilized eggs. Total RNA was extracted from sperm using the RNA Blue reagent according to the manufacturer's instructions (Top-Bio, Czech Republic). Inactivation test of some heat shock proteins was done by geldanamycin solution that is able to block these proteins. All chemicals were supplied by Sigma-Aldrich (Sigma-Aldrich s.r.o. Prague, Pobrezni 46, Czech Republic).

**Injecting eggs.** Total RNA, homogenized protein or geldanamycin were dissolved in injection buffer and approximately 0.2–1 fmol of RNA, 0.05 μg proteins or 0.5μg geldanamycin (ca 0.3 μl of injection buffer) were injected into the ventral side of the posterior domain of *Ephestia* embryos (Fig 4), similar to *Drosophila* or *Chymomyza* [27]. One-three day old fertilized eggs were injected. The control was performed only with the buffer.

## The influence of a diet on epigenetic effect

Larvae (*sa*) were fed milled wheat grains supplemented with a small amount of dried yeast (optimal food for ontogenetic development). In two other experiments, larvae were fed wheat with added NaCl or LiCl to 1.1 on 1 mg of food (additives affecting suitable food). Finally, the larvae were fed on plain flour only (unsuitable food for proper ontogenetic development).

## Duration of the epigenetic phenomenon

Over 10–20 lines in each generation (always descendants of one pair) carrying epigenetic information were monitored for forty generations. We counted a proportion of epigenetic

trait (*sa*<sup>WT</sup>) and non-epigenetic trait (*sa*) for individuals in F1–F3, F5 and F12 generations. F4, F6–F11 and F13–F40 generations were just observed without precise counting of *sa* and *sa*<sup>WT</sup> phenotype.

## Statistical analyses

We counted the number of offspring with short antennae (*sa*) and long antennae (*sa*<sup>WT</sup>) in each moth's pair (clutch) and calculated the proportion of wild *sa*<sup>WT</sup> offspring (imagoes) out of all offspring in the clutch. To find out the molecular basis of the inheritance of the epigenetic phenomenon, we calculated the average proportion of *sa*<sup>WT</sup> phenotype for all eight factors injected to the eggs (buffer, geldanamycin, sperm, accessory gland, homogenized fractions of sperm, homogenized spermatophore sac, sperm and RNase, and total RNA, see above) and compared them with non-parametric Kruskal-Wallis ANOVA followed by Dunn post-hoc tests. Ordinary one-way ANOVA cannot be used as the data does not met ANOVA assumptions (normality by groups and homogeneity of variances). P-values of post-hoc tests were adjusted by Benjamini-Hochberg correction to control the false discovery rate. The same statistical tests were performed to evaluate the influence of four types of diet (flour, wheat with LiCl, wheat with NaCl, wheat grains) on epigenetic effect.

To assess the duration of the epigenetic memory, we computed the number of extremely reversed and non-reversed clutches and calculated the proportion of extremely reversed clutches in F1–F3, F5 and F12 generations of moths and compared them with goodness-of-fit chi-square test. A clutch was classified as extremely reversed if the percentage of reversed individuals in the clutch was greater than 12.6%. This threshold was defined as the non-outlier maximum over clutches of all generations. All tests were performed in R version 4.0.2 (2020 The R Foundation for Statistical Computing).

## Results

### The molecular basis of epigenetic phenomenon

Descriptive statistics of proportion of *sa*<sup>WT</sup> phenotype in the offspring of moth's pair for eight substances injected to eggs (additives) are summarized in Table 1 and Fig 5. According to Kruskal-Wallis ANOVA, the proportion of *sa*<sup>WT</sup> offspring varies among the eight additives ($P<0.001$). Dunn's post-hoc tests (Table 2) show several significant pairwise differences. In general, proportion of *sa*<sup>WT</sup> phenotype differs only between egg additives with and without RNA content. By contrast, post-hoc tests suggest that proportion of *sa*<sup>WT</sup> imagoes is similar in all four additives without RNA (buffer, product of accessory gland, homogenized spermatophore sac, and sperm and RNase) and in all four additives with RNA (geldanamycin, sperm, homogenized fractions of sperm, and total RNA).

When four additives with RNA and four without RNA content are combined together into two groups, differences between means of percentage of *sa*<sup>WT</sup> phenotype is highly significant (Kruskal-Wallis ANOVA, $P<0.001$). It means that injection of total RNA isolated from *sa*<sup>WT</sup> sperm into fertilized eggs (*sa* x *sa*) produced a significantly higher percentage of offspring with *sa*<sup>WT</sup> long-antennae.

### Duration of the epigenetic phenomenon

During the monitoring of generations of single-pair cultures, we found that there is a reversion to the initial mutant phenotype by some individuals. Proportion of reverted individuals is usually 0–6% per generation and pair. Rarely, the offspring of some pairs reverted in large number (90% and more). The non-outlier maximum of proportion of reverted individuals for clutches

**Table 1.** Mean, median, and standard deviation (SD) of proportion of $sa^{WT}$ imagoes of each moth's pair by three different types of interventions.

| Type of intervention | Number of imagoes | Number of moth's pairs | Mean | Median | SD |
|---|---|---|---|---|---|
| Substance injected to eggs | | | | | |
| Buffer | 390 | 7 | 27.2 | 11.8 | 30.2 |
| Geldanamycin | 141 | 5 | 48.5 | 51.7 | 14.7 |
| Sperm | 636 | 17 | 52.1 | 51.7 | 29.0 |
| Product of accessory gland | 187 | 6 | 18.6 | 18.6 | 11.8 |
| Homogenized fractions of sperm | 456 | 7 | 43.9 | 52.2 | 26.4 |
| Homogenized spermatophore sac | 152 | 5 | 8.0 | 7.7 | 2.1 |
| Sperm and RNase | 164 | 10 | 22.4 | 17.9 | 18.9 |
| Total RNA | 140 | 7 | 63.2 | 66.7 | 24.6 |
| Substance by RNA content | | | | | |
| Yes | 1373 | 36 | 52.1 | 51.9 | 25.9 |
| No | 893 | 28 | 20.2 | 20.211.9 | 19.8 |
| Food medium | | | | | |
| Flour | 340 | 8 | 90.2 | 92.1 | 5.6 |
| LiCl | 485 | 11 | 85.7 | 93.1 | 16.3 |
| NaCl | 244 | 12 | 89.9 | 88.7 | 7.3 |
| Wheat grains | 1243 | 15 | 11.0 | 10.0 | 6.8 |

from all five generations monitored in details 12.6%, which for us is the point at which a significant number of individuals were reverted (see Methods). Numbers of extremely reverted clutches (proportion of reverted individuals greater than 12.6%) and non-reverted clutches (proportion of reverted individuals less than 12.6%) by generation are given in Fig 6. The results of the goodness-of-fit test show that the proportions of extremely reverted clutches do

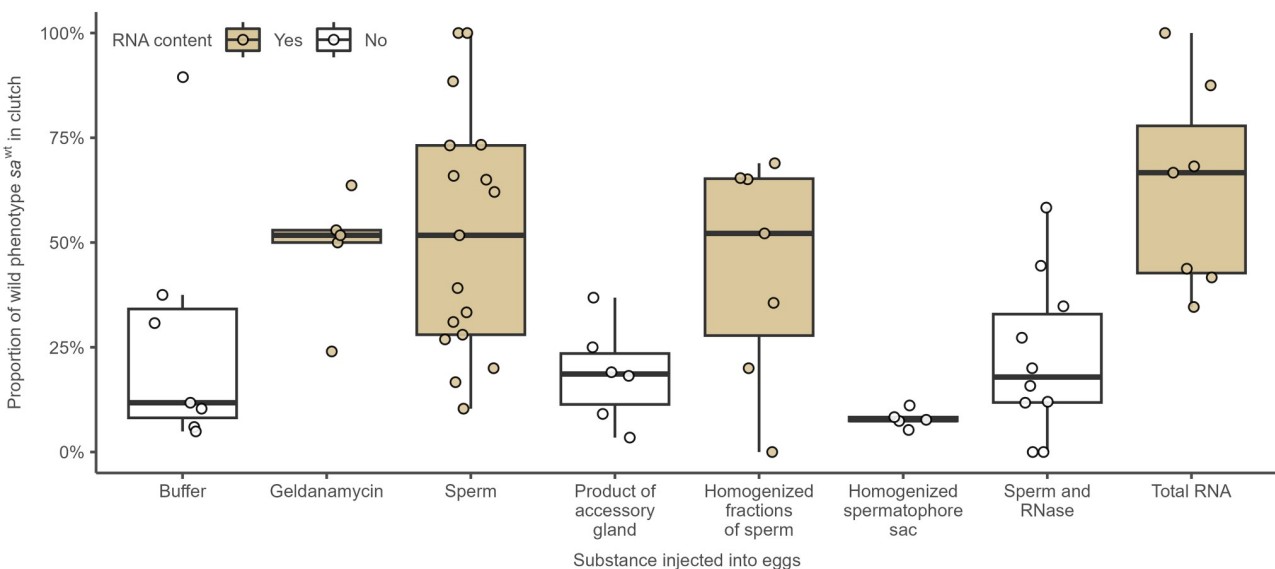

**Fig 5. Boxplots of proportion of $sa^{WT}$ offspring between eight different types of additives injected into the eggs.** The central thick line is median, box corresponds to lower and upper quartiles and whiskers correspond to non-outlier minimum and maximum. Circles are raw values of Y variable in each clutch. Grey and white fill corresponds to additives with and without RNA content, respectively.

**Table 2. Dunn's pairwise comparison of proportion of saWT imagoes between eight types of substances injected to eggs.** P-values are adjusted with Benjamini-Hochberg correction.

| | Buffer | Geldan# | Sperm# | Product of accessory gland | Homogen. fractions of sperm# | Homogen. sperm. sac | Sperm and RNase |
|---|---|---|---|---|---|---|---|
| Geldanamycin# | 0.218 | | | | | | |
| Sperm# | 0.099 | 0.959 | | | | | |
| Product of accessory gland | 0.788 | 0.146 | 0.041* | | | | |
| Homogenized fractions of sperm# | 0.294 | 0.814 | 0.783 | 0.175 | | | |
| Homogenized spermatophore sac | 0.296 | 0.040* | 0.009** | 0.500 | 0.041* | | |
| Sperm and RNase | 0.925 | 0.168 | 0.041* | 0.814 | 0.206 | 0.296 | |
| Total RNA# | 0.041* | 0.596 | 0.500 | 0.032* | 0.378 | 0.008** | 0.032* |

\* Statistically significant at 0.05 level,

\*\* at 0.01 level

\# Substance with RNA content

not differ among F1, F2, F3, F5, and F12 generations (chi-square = 2.6, df = 4, P = 0.64). Five cultures were observed in subsequent generations and the long antennae epigenetic effect continued to 40th generation.

## Medium (the effect of salts and poor nutrition)

The proportion of phenotype changes ($sa$ on $sa^{WT}$) for four different food medium is shown in Table 1 and Fig 7. Kruskal-Wallis ANOVA indicates that differences among group means were significant (P<0.001). Dunn's post-hoc tests further show that three food medium (flour, LiCl, and NaCl) had same proportion of phenotypically altered individuals and all these three groups differ from control (individuals that were fed with wheat grains) (P values < 0.001).

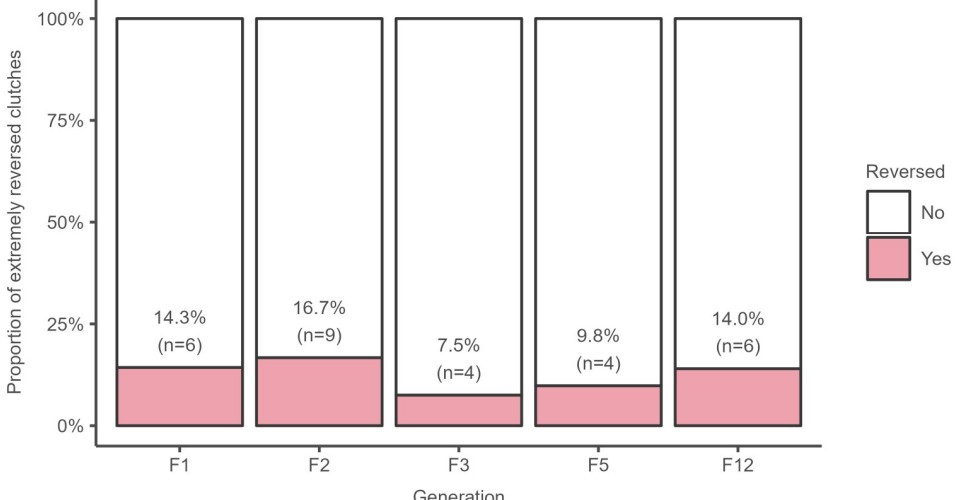

**Fig 6. Absolute and relative number of extremely reversed clutches by generation.** Extremely reversed clutch was defined as a clutch with a percentage of reversed individuals higher than 12.6% (overall non-outlier maximum). The total number of clutches (n) is given for each generation.

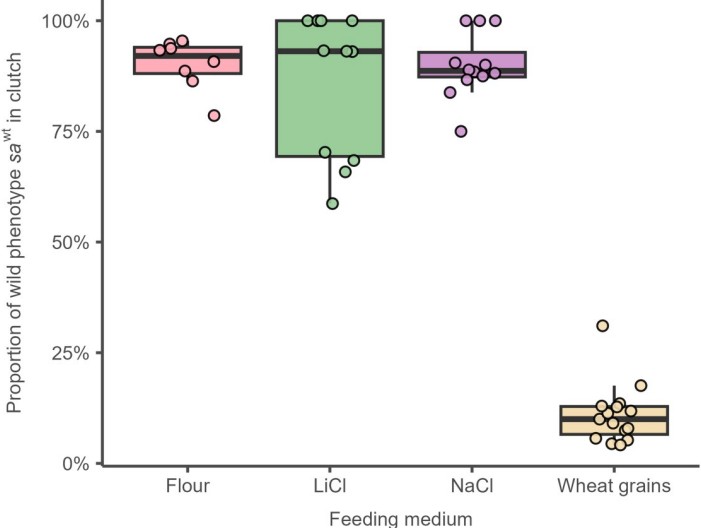

**Fig 7. Boxplots of percentage of $sa^{WT}$ offspring between four different food medium.** The central thick line is median, box corresponds to lower and upper quartiles and whiskers correspond to non-outlier minimum and maximum. Circles are raw values of Y variable in each clutch.

## Discussion

### Epigenetic expression of morphological mutations

To the best of our knowledge, the effect of RNAs in *E. kuehniella* generations was observed for the first time here. Mutant short antennae changed to the original long wild type due to the influence of certain RNA or some environmental factors. The change to the wild-type phenotype appears permanent, but occasionally the original mutant form returns. We observed that reversion to the original mutant phenotype occurred only in a small proportion of offspring and rarely in the entire population (Table 1, Figs 5 and 6).

A similar phenomenon has been observed in mammals and nematodes. Rassoulzadegan et al. [24] reported a modification of the mouse *Kit* gene in the progeny of heterozygotes with the null mutant *Kit*[tm1Alf] (a *lacZ* insertion). Wild male mice mated with female mutant (heterozygous mouse *Kit*[tm1Alf]) have offspring with homozygous wild genes and white spots like the mutant female even though they have no allele for these spots. In homozygous offspring of the wild-type genotype, the white spots characteristic of *Kit* mutant animals persist to varying degrees [24]. Large amount of an aberrant RNA is produced from the mutant gene (*Kit*[tm1Alf]), consequently is accumulated in sperm, and thus is transmitted to the embryo. Presence of the aberrant RNA silences the activity of the Kit wild-type gene then so the animals have white spots, even if they do not carry the corresponding mutant gene [28]. Epigenetic changes may disappear at the beginning of each new generation. However, in some cases, epigenetic information can be transmitted from parent to progeny (multigenerational epigenetic inheritance) [29]. A particularly notable example of this type of epigenetic inheritance is double-stranded RNA-mediated gene silencing (RNAi) in *C. elegans*. This phenotype caused by RNAi could be inherited for more than five generations [1].

### Duration of the epigenetic phenomenon

The transmission of epigenetic traits across multiple generations is not rare [30]. In the case of *E. kuehniella*, our results revealed that the epigenetic effect persisted for at least 40 generations.

It is probably one of the longest durations of epigenetic influence ever observed in an insect. It is likely that epigenetic effect would continue even further to next generations because the phenotypes with the suppressed mutation still persisted. It seemed to be a permanent phenomenon because there was no statistical change in the proportion of extreme clutches among F1, F2, F3, F5, and F12 generations. The effect of 40 generations in sexually reproducing animals has been rarely demonstrated. An example of a similar phenomenon in insects is the *trans*-silencing effect (TSE), involved in *P*-transposable-element repression in the germ line, which transmits the acquired silencing capacity for 50 generations [31].

The observed epigenetic phenomenon could be related not only to piRNAs, but also to transposable elements (TE). In *D. melanogaster*, paramutation is correlated with transmission of PIWI-Interacting RNAs (piRNAs), a class of small non-coding RNAs that repress mobile DNA in the germline [32]. In insects, most transposable elements silencing in the germline is achieved by secondary piRNAs that are produced by a feed-forward loop (the ping-pong cycle), which requires the piRNA-directed cleavage of two types of RNAs: mRNAs of functional euchromatic TEs and heterochromatic transcripts that contain defective TE sequences. This capacity to produce heterochromatic-only secondary piRNAs is partially transmitted through generations via maternal piRNAs [33]. Epigenetic interactions labeled in this case as paramutation are interactions between two alleles at a given locus, in which one allele induces a heritable modification of the other without modifying the DNA sequence. In *D. melanogaster* it was discovered that clusters of *P*-element-derived transgenes that induce a strong *trans*-silencing effect (TSE) can convert other homologous transgene clusters incapable of TSE into strong silencers, which in turn transmit the acquired silencing capacity. The paramutated cluster is converted into a stable, strong piRNA-producing locus and becomes fully paramutagenic itself [31]. From the point of view of this study, it is essential that the epigenetic effect (paramutation) is transmitted for the whole 50 generations, which is analogous to the suppression effect of the *sa* mutation. But here it is a matter of transmission only through the maternal line, not the paternal line as in our case.

Highly stable long-term epigenetic silencing effect lasting at least 20 generations can be triggered in *C. elegans* by piRNA. Once established, this long-term memory becomes independent of the piRNA trigger, but remains dependent on the nuclear RNAi/chromatin pathway [34]. The mechanisms associated with piRNA are probably common to all animals, at least in some basic form. It is still a question, how frequent their transgenerational effect is, but in *D. melanogaster* it has already been proven to act after 50 generations [31]. Insects benefit more from the pronounced variability of the progeny, because they produce much larger numbers of offspring and are physiologically more influenced by environmental conditions than mammals [35]. Epigenetic memory in mice lasts only six generations [24], in voles is the memory documented only to F3 generation [36]. However, the situation is different in insects. In *D. melanogaster*, the most epigenetic changes were recorded for 11–13 generations [37, 38]. For instance the studied epigenetic trait (developed antennae) is involved in finding of sexual partner [14] and lasts for many generations. It was not observed in mammals. Such long lasting epigenetic effect could be considered potentially adaptive [23, 39]. It is possible that epigenetic phenomena could have evolutionary consequences that increase variability in offspring [30]. We hypothesize that the long-lasting epigenetic effect observed in our experiments could indicate that there is a simpler regulatory mechanism involved in insect cells compared to mammals.

## The molecular basis of epigenetic phenomenon

The male flour moth does not only deposit sperm into the female, but a spermatophore that contains also other components. As mentioned in Methods, these components were separated

and injected separately into the fertilized eggs. Other components without RNA content (product of accessory glands, spermatophore sac, homogenized sperm denatured with Ribonuclease) separated from male sperm did not have analogous impact like components with RNA. The total RNA differed from all additives without RNA. The influence of geldanamycin comparable to the RNA injection was even stronger than expected, which probably confirming that the epigenetic effect was due to the small RNA (Tables 1 and 2, Fig 5). Geldanamycin inhibits heat shock protein 90 (Hsp90) [40], which is a molecular chaperone essential for activating many signaling proteins in the eukaryotic cell [41]. The link between small RNA within Argonaute proteins (Argonaute proteins bind different classes of small RNAs) and Hsp90 has been demonstrated [24]. The loading of siRNA duplexes into *Drosophila* Ago2 requires the Dicer-2–R2D2 heterodimer and the Hsc70/Hsp90 chaperone machinery. In the absence of the chaperone machinery, an siRNA bound to Dicer-2–R2D2 associates with Ago2 only transiently [42].

As the RNA is not degraded and continues to act in the cells, there might be a relationship with RNAs generating interference RNA (RNAi) [28]. In this process, RNA methyltransferases seem to be also essential. Research in animal models has shown that RNAs can be inherited and that RNA methyltransferases can be important for the transmission and expression of modified phenotypes in the next generation [15, 43]. Furthermore, RNA methyltransferases increase the stability of small RNA as cytosine-5 methylation [15, 44]. Similar phenomenon conditioned by miRNAs found in lepidopterans has been observed for example in plants where a temperature-dependent epigenetic memory from the time of embryo development expresses in norway spruce (Pinaceae). This epigenetic machinery influences the timing bud phenology [45]. The understanding of the role of small RNAs continues to deepen in insects also playing a role in gametogenesis. The small RNAs may play a fundamental role in honey bee gametogenesis and reproduction and provide a plausible mechanism for parent-of-origin effects on gene expression and reproductive physiology [46].

The epigenetic capabilities of piRNAs in intergenerational transmission through the male germline have been noted [47]. Although there have been many examples of sRNA-mediated epigenetic inheritance in *C. elegans*, other organisms which do not have RNA-dependent RNA polymerases (RdRP), do not seem to exhibit a similar repertoire for inheriting various stress induced responses [4]. The results of this study suggest that similar mechanisms could be at work in insects. Perhaps methylation plays a role at this level as well, and regulation by sRNA is not entirely specific. Using a novel approach, which can differentiate between primary (inducer) and secondary (amplified) sRNAs, it was shown that initiation of heritable RNA-directed DNA methylation (RdDM) does not require complete sequence complementarity between the sRNAs and their nuclear target sequences [48].

Many studies have demonstrated that epigenetic molecular mechanisms, including DNA methylation and histone modification, not only regulate the expression of protein-encoding genes, but also miRNAs [46]. Conversely, another subset of miRNAs controls the expression of important epigenetic regulators, including DNA methyltransferases, histone deacetylases, and polycomb group genes [48]. This complicated network of feedback between miRNAs and epigenetic pathways appears to form an epigenetics-miRNA regulatory circuit, and to organize the whole gene expression profile [49, 50].

But this feedback network does not always work, as evidenced by the recorded reversion of the mutant phenotype, which occurs spontaneously in some, or rarely in all, offspring of the studied flour moths (Fig 6). We suppose that it means the diminishing of epigenetic influence on the basis of gradual decrease of sRNAs (maybe piRNA) during ontogeny under a particular treshold. So, an individual posses epigenetic trait, but all its tissues contain no sRNAs causing epigenetic effect any more. It could occur during ontogeny of some individuals, whereas others

could retain the same concentration of sRNAs. Occasionally, in rare cases could some small RNAs disappear only in the parental tissues where eggs and sperm develop, while the individuals and their other tissues display epigenetic phenotype. The offspring then do not display epigenetic alterations of its phenotype.

## Environmental factors induced an epigenetic effect

The epigenetic phenomenon described in the current and previous study [25] is peculiar in that it was induced by a change in temperature during the development of individuals. The influence of other environmental factors on the induction of an epigenetic response was also detected, whether the *E. kuehniella* were fed either on flour or on wheat with LiCl or NaCl additives. Although we found no statistical differences between LiCl, NaCl, and flour in the proportion of $sa^{WT}$ individuals, all three investigated groups differed from the spontaneous onset of $sa^{WT}$ in the optimal diet (milled wheat grains supplemented with a small amount of dried yeast). Similarly, the effect of injections into the eggs was manifested (Table 1, Fig 5). However, the injection with buffer only, which served as the control, was able to elicit an epigenetic response (to some extent) because the injection itself is a major intrusion into the developing egg (Tables 1 and 2, Fig 5). The developing embryo was mostly killed immediately after injection (the ratio of mortality was not recorded).

Phenotypic plasticity is an ubiquitous process found in all living organisms. Polyphenism is an extreme case of phenotypic plasticity which shares a common scheme in insects such as honeybees, locusts or aphids. Climate change can modulate the environmental stimuli triggering polyphenisms, and/or some epigenetics marks, thus modifying on the short and long terms the discrete phenotype proportions within populations [51]. A similar effect of temperature such as the subject of study was observed in *C. elegans* (Nematoda). It was demonstrated that a subset of synMuv B mutants ectopically misexpress germline-specific P-granule proteins in their somatic cells, suggesting a failure to properly orchestrate a soma/germline fate decision [52]. A majority of the SynMuv B mutants grown at high temperature were irreversibly arrested at the L1 stage. Somatic expression of germline genes is enhanced at elevated temperature, leading to developmentally compromised somatic cells and arrest of newly hatched larvae. High temperature arrest is accompanied by upregulation of many genes characteristic of the germ line, including genes encoding components of the synaptonemal complex and other meiosis proteins [52]. Unfortunately, the link to small RNAs is unclear.

## Possible association of epigenetic change with sirtuin genes

The most common manifestation of epigenetic change can be seen in the sirtuin genes [53–55]. The similarity with our phenomenon is primarily in the long-term inheritance genotype, which is occasionally interrupted for reasons which remain unclear. It is still unanswered why this mechanism is triggered by a change in temperature. The SIR2 gene, for example, could activate the production of small RNAs. White–opaque switching in the human fungal pathogen *Candida albicans*, results from the alternation between two distinct types of cells [53, 54]. Switching is probably caused by the SIR2 (silent information regulator) gene, which seems to be important for phenotypic switching [55]. Silent Information Regulator (SIR) proteins are involved in regulating gene expression and some SIR family members are conserved from yeast to humans [56]. SIR genes have many functions. Sirtuins are evolutionary conserved NAD(+)-dependent acetyl-lysine deacetylases and ADP ribosyltransferases dual-function enzymes involved in the regulation of metabolism and lifespan [57]. *Sirtuins are hypothesized to play a key role in an organism's response to stresses (such as heat or starvation)*. A calorie restriction turns on a gene called PNC1, which produces an enzyme that rids cells of

nicotinamide, a small molecule similar to vitamin B3 that normally represses Sir2. The gen PNC1 is also stimulated by other mild stressors known to extend yeast life span, such as increased temperature or excessive amounts of salt [58].

Based on this similarity, we speculate that SIR proteins in addition to various known functions (e.g., silence genes–see Ralser et al. [59]) might also catalyse the formation of small double-stranded RNA, according to damaged genes to be silenced. SIRT regulation is multifaceted, but not yet considered to be associated with RNA. We present the hypothesis that insects can initiate the creation of RNA against harmful genes. SIR genes linked with small RNAs have been reported. It was shown that small interfering RNA-mediated SIRT7 knockdown leads to reduced levels of RNA Pol I protein, but not messenger RNA, which was confirmed in diverse cell types [60].

### What do the reversions to the original mutant type mean?

The studied epigenetic phenomenon affected the majority of the flour moth population, but not all individuals. A small number of individuals often revert to the original mutated type (short antennae, Fig 6). It is possible that it is a mechanism that increases the variability of the population and thus increases the chance of the species to survive. This effect is therefore evolutionarily significant. It is possible that switching between states occurs because certain small RNA in the fertilized eggs is lower than the critical concentration required. However, the mechanism can be far more complex.

The epigenetic inheritance we discovered is probably an adaptive property of the organism. Especially when we consider that flour moths with normal antennae mate better than normal *sa* mutants [25]. Adult flour moths only live about 15 days, and their sole purpose in the adult state is to mate and lay eggs, and they do not even take food. Our work supports the idea that many epigenetic mechanisms are related to hereditary phenotypic plasticity, as one of the evolutionary mechanisms [20].

### Conclusion

The epigenetic phenomenon described in the current and previous study [25] (the phenotypic reversion from the mutant short antennae to the wild-type long antennae) is peculiar in that it was induced by a change in temperature during the development of individuals. The effect is apparently caused by specific RNAs that are formed during environmental stress. A connection with heat shock protein 90 (Hsp90) and Argonaute proteins is also possible. The epigenetic effect is very stable and has been observed for 40 generations. Occasionally, however, the epigenetic effect disappeared. We suppose that it means the diminishing of epigenetic influence on the basis of gradual decrease of sRNAs (maybe piRNA) during ontogeny under a particular treshold. Our work supports the idea that many epigenetic mechanisms are related to hereditary phenotypic plasticity as one of the evolutionary mechanisms.

### Supporting information

**S1 Data.**
(XLS)

### Acknowledgments

The research and findings reported in this study were conducted and delivered before funding and research support was terminated in 2004. Unfortunately, it is published only now.

## Author Contributions

**Conceptualization:** Jaroslav Pavelka.

**Data curation:** Jaroslav Pavelka, Patrik Galeta.

**Formal analysis:** Simona Poláková, Věra Pavelková, Patrik Galeta.

**Investigation:** Jaroslav Pavelka.

**Methodology:** Jaroslav Pavelka, Patrik Galeta.

**Project administration:** Jaroslav Pavelka.

**Software:** Patrik Galeta.

**Supervision:** Věra Pavelková.

**Visualization:** Patrik Galeta.

**Writing – original draft:** Jaroslav Pavelka.

**Writing – review & editing:** Jaroslav Pavelka.

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
