## [Decision Letter · Decision Letter 0]

28 Dec 2022

PONE-D-22-29568An epigenetic change in a moth is generated by temperature and transmitted to many subsequent generations mediated by RNAPLOS ONE

Dear Dr. Pavelka,

Thank you for submitting your manuscript to PLOS ONE. After careful consideration, we feel that it has merit but does not fully meet PLOS ONE’s publication criteria as it currently stands. Therefore, we invite you to submit a revised version of the manuscript that addresses the points raised during the review process. Both Reviewers provided a number of suggestions that should be considered, many of which focus on improving the overall clarity of the manuscript. In particular, I agree with the comments from Reviewer 1 that more methodological details in the Results section, the inclusion of schematics describing what was done, and specific photos of the phenotypes would alleviate much of the confusion. Input from a native speaking English colleague on the more complicated sections (e.g. the first Results section), which are unnecessarily dense and potentially impede reader comprehension, and more thorough copy editing (e.g. Figure 3 is referenced before Figure 2 in the Results) would also make the manuscript more accessible. Lastly, the inclusion of page and line numbers would greatly facilitate the review process.

We look forward to receiving your revised manuscript.

Kind regards,

J Joe Hull, Ph.D.

Academic Editor

PLOS ONE

Journal Requirements:

https://www.tdx.cat/handle/10803/668338

https://ksn.or.kr/upload/journal/701525348/2017/701525348_2017_1514277222.pdf

In your revision ensure you cite all your sources (including your own works), and quote or rephrase any duplicated text outside the methods section. Further consideration is dependent on these concerns being addressed.

Reviewers' comments:

Reviewer's Responses to Questions

**Comments to the Author**

1. Is the manuscript technically sound, and do the data support the conclusions?

Reviewer #1: Yes

Reviewer #2: Partly

2. Has the statistical analysis been performed appropriately and rigorously? 

Reviewer #1: Yes

Reviewer #2: Yes

3. Have the authors made all data underlying the findings in their manuscript fully available?

Reviewer #1: Yes

Reviewer #2: Yes

4. Is the manuscript presented in an intelligible fashion and written in standard English?

Reviewer #1: Yes

Reviewer #2: No

5. Review Comments to the Author

Reviewer #1: Review of PLoS One manuscript “An epigenetic change in a moth is generated by temperature and transmitted to many subsequent generations mediated by RNA” by Pavelka et al.

Summary

The research team investigates epigenetic inheritance of a antenna phenotype in a moth. They use experimental techniques to determine the nature of the factors that affect this ‘paramutation’. They suggest that RNA may be involved in the epigenetic inheritance of this phenotype.

The study epigenetics is important. And I appreciate the experimental approach applied in this study. The experimental outcomes do provide information on the nature of inheritance of this phenotype. And the results are potentially interesting.

But I was confused by the results (as were the authors, no doubt) which made the nature of the paramutation a bit hard to explain. That is, the results in Fig 1 and Table 1 are difficult to understand. Extracts that contained RNA had higher sawt phenotypes, which is interesting. But why did the buffer negative control show any sawt phenotypes at all? Why doesn’t the accessory gland (which also contains RNA?) not produce an effect? Why wouldn’t ‘homogenized fractions of sperm’ contain RNA that would lead to the same outcome as ‘sperm’? These are difficult results to explain given the nature of the analyses.

Overall, I view the study favorably from the standpoint of presenting novel results. But it was still hard to know exactly what was going on in this system. So I have few comments on this manuscript and found the study to be of interest, albeit presenting unclear conclusions.

Comments

The overall research would be aided if the research team new exactly which gene (or gene products) were involved in causing the mutation. That is, the sa mutation is said to be a simple recessive. But it would be very important to know exactly which gene, and what type of DNA mutation, is involved in generating the initial phenotype. I realize this is beyond the scope of this article, but it should be addressed.

I found the Introduction to be well written. I liked the description of examples of previous epigenetic RNA transmission (although some of this perhaps could go in the Discussion). Because epigenetic inheritance is weird, it is important for the research team to provide ample support for their speculative hypothesis.

I don’t know if PLoS requires Results to come before Methods. But, in this case, it is problematic. I would strongly urge the authors to put their Methods before the Results. This is a paper about Methods and so it is confusing, and a bit silly, to essentially tell the reader that they have to read the Methods first in the Results section. Why not just put the Methods first?

The Methods got quite confusing. The authors mention “prolonged antennae (saWT) with changed phenotype”. But they don’t explain exactly what that is. They say “This mutation was turned out at 25°C ± 1°C”. But I don’t know what ‘turned out’ means here. I assume this means reversion.

The authors go on to say that “the male spermatophore (from saWT male) was analyzed and its individual parts were then injected into the previously fertilized eggs”. What does it mean “analyzed”? What are the “individual parts” of a sperm? How do you inject these “individual parts” into an egg? More explanation is needed.

I think the paper would be greatly aided by showing more photos of the experimental procedures. For example, please include photos of the different antennae phenotypes. Also, include photos of the ‘parts’ of the sperm if possible. And a photo of egg injection would also be helpful.

Do Table 1 and Figure 1 present the same data? If so, it is confusing and, therefore, the authors probably need not present both.

Table 2 shows some kind of comparison of significance of differences. Such information is useful. But these tests are usually displayed using more conventional techniques whereby groups of means that differ from each other are indicated by different letters in a figure of means (such as Fig 1). This would be clearer as a display item.

Reviewer #2: The manuscript titled: “An epigenetic change in a moth is generated by temperature and transmitted to many subsequent generations mediated by RNA” presents interesting results regarding transgenerational epigenetic inheritance triggered by different stressors early in the development of the flour moth (Ephestia kuehniella). Differences triggered by additive stressor, plus RNA content or not, injected in E. kuehniella eggs where reflected in antennae length phenotypic differences (sa and saWT) for over 20 generations.

These findings are worth to be communicated and published. Nonetheless, this version sent to review it is too preliminary regarding theoretical background revision, writing and grammar.

Several typos and English writing corrections where highlighted in the attached pdf. Thus, I will focus on major necessary changes that must to be implemented before this work can be accepted for publication.

Abstract

Several grammar issues shown in the pdf with suggestions attached. Besides this:

1-When you comment that this effect last “tens of generations” please be more specific and explain how many generations.

2- When you present the “wild phenotype” please describe in few words.

Author summary

Also, several changes suggested in grammar and writing.

Introduction

Several grammar issues shown in the pdf with suggestions attached. In addition:

1-Explain transgenerational epigenetic inheritance. For this, I recommend you to read and cite:

Skinner, MK (2014). Environmental stress and epigenetic transgenerational inheritance. BMC Medicine

Yang, C; Zeng, QX; (...); Duan, YG (2022) Role of small RNAs harbored by sperm in embryonic development and offspring phenotype. Andrology

Rothi, MH and Greer, EL (2022) From correlation to causation: The new frontier of transgenerational epigenetic inheritance. Bioasays.

Roschdi, S; Yan, J; (...); Butcher, SE (2022) An atypical RNA quadruplex marks RNAs as vectors for gene silencing. Nature Structural and Molecular Biology.

2- When you describe that: “acquired traits can be memorized' in the sperm as epigenetic” changes or alterations. Please after this, you must briefly explain main epigenetic molecular mechanisms with special attention on the role of sRNAs.

Here use the following reference:

Silva W., Otto S.P. & Immler S. (2021). Evolution of plasticity in production and transgenerational inheritance of small RNAs under dynamic environmental conditions. PLoS Genet.

3- After “(e.g., temperature, LiCl in food); and 2) under certain conditions”. Please explain these conditions and cite references.

4- Following “of antennae in moths of both sexes.” Add cite.

5- I’m not sure what you mean with this phrase. Regarding the study case? Please be more precise.

6- When you say “It is still an unanswered question what exactly causes the transmission of epigenetic information to offspring.” I do not understand what you mean.

If you are referring in general to the effects of Epigenetic Molecular Mechanisms on transgenerational epigenetic inheritance, these links are more or less well known and expanding. You should review further this topic and be precise what is the novelty of this work.

7- From “Even though this type of inheritance.” to “could survive to at least 20 generations [16]” you state commentaries about the study case and its relevance that I recommend you to move to the discussion. In intro you must finish describing the study system and the goals of the study with less divergent paths of thoughts than a discussion of a given topic. I’m confident that after you read and update on the available works on sRNA role you will be able to re-write this introduction with ease.

Check:

Colicchio J., Kelly J., Hileman L. (2021) Mimulus sRNAs Are Wound Responsive and Associated with Transgenerationally Plastic Genes but Rarely Both. Int J Mol Sci.

Fei, Y; Nyiko, T and Molnar, A (2021) Non-perfectly matching small RNAs can induce stable and heritable epigenetic modifications and can be used as molecular markers to trace the origin and fate of silencing RNAs. Nucleic Acids Research.

Watson, OT; Buchmann, G; (...); Ashe, A (2022).Abundant small RNAs in the reproductive tissues and eggs of the honey bee, Apis mellifera. BMC genomics.

8- By the end of Intro you must present your system properly and describe the case you are going to study. This especially important for PLOS format where methodological information is at the end of the article. Please explain enough about your moths, antenna phenotype alternatives and what it is know so far. Present your current work. Then give hits about what you found.

Intro needs re-writing.

Results

Mainly minor writing issues. See pdf attached.

Discussion

Beside English proofreading problems:

1-In general when you comment on your results please add a reference to your material (Figure) or (Table).

2- When you mention that RNA-related epigenetic changes lasted “for 20 (or 40) generations” Explain when in each case.

3- The phrase starting with “Unlike the other components.” It is not clear, please rephrase it.

4- Iwasaki et al 2020 is not in PLOS citing format.

5- When you compare with C. elegans, please explain further, it is too brief.

6- After “Insects benefit more from the pronounced variability of the progeny because they produce much larger numbers of offspring and are physiologically more influenced by environmental conditions than mammals” Please provide references for your statement, as this is an evolutionary comparison. I’m not sure what you say here it is demonstrated so far.

7- After “However, the situation is different in insects” Please provide examples and cite.

8- Following “histone deacetylases, and polycomb group genes” add supporting references.

9-After “The most common manifestation of epigenetic change it can be seen in the sirtuin genes”. Add references for this phrase.

Materials and Methods

Several typos, English errors and writing problems, suggestions in the pdf. Also:

1-Animals and breeding: Please present and describe short antennae (sa) and prolonged antennae (sa WT ) in the section where you describe the mutant phenotype.

2- Isolation of separate parts of spermatophore: When you describe that: “Immediately after the copulation ended (sa WT male and female), the female was dissected”. Please add what you aim with this. What do you extracted from females? Explain a little bit more.

3- Statistical Analyses: When you mention that: “We have tried to determine the point at which a significant number of individuals in populations”. The phrase is not clear. Please re-write.

One problem I faced review this ms is the lack of page numbers and line numbers. Please include this in your future ms versions.

Same comments sent to editors.

6. PLOS authors have the option to publish the peer review history of their article (what does this mean?). If published, this will include your full peer review and any attached files.

Reviewer #1: No

Reviewer #2: **Yes: **Cristian Villagra

---

## [Author Response · Author response to Decision Letter 0]

1 Apr 2023

The manuscript has been returned for revision, I am sending the revised version.

---

## [Decision Letter · Decision Letter 1]

12 Jun 2023

PONE-D-22-29568R1An epigenetic change in a moth is generated by temperature and transmitted to many subsequent generations mediated by RNAPLOS ONE

Dear Dr. Pavelka,

My apologies for the delay with this decision letter, the process was lengthened when Reviewer 1 became unexpectedly unavailable. Although your consideration for the Reviewer comments in the revised manuscript are appreciated, we feel that further revision is needed for the manuscript to fully meet PLOS ONE’s publication criteria. Therefore, we invite you to submit a revised version of the manuscript that addresses the points raised during the review process. Reviewer 3, in particular, provided a number of comments and suggestions that should be considered about making the Methods more accessible to a wider audience and focusing the Discussion. Reviewer 2 likewise provided a number of useful suggestions (detailed below as well as the downloadable edited pdf). Lastly, I am in full agreement with Reviewer 2 regarding the changes that must be made to the Acknowledgements.

We look forward to receiving your revised manuscript.

Kind regards,

J Joe Hull, Ph.D.

Academic Editor

PLOS ONE

Reviewers' comments:

Reviewer's Responses to Questions

**Comments to the Author**

1. If the authors have adequately addressed your comments raised in a previous round of review and you feel that this manuscript is now acceptable for publication, you may indicate that here to bypass the “Comments to the Author” section, enter your conflict of interest statement in the “Confidential to Editor” section, and submit your "Accept" recommendation.

Reviewer #2: All comments have been addressed

Reviewer #3: (No Response)

2. Is the manuscript technically sound, and do the data support the conclusions?

Reviewer #2: Partly

Reviewer #3: No

3. Has the statistical analysis been performed appropriately and rigorously? 

Reviewer #2: N/A

Reviewer #3: No

4. Have the authors made all data underlying the findings in their manuscript fully available?

Reviewer #2: Yes

Reviewer #3: Yes

5. Is the manuscript presented in an intelligible fashion and written in standard English?

Reviewer #2: No

Reviewer #3: No

6. Review Comments to the Author

Reviewer #2: The work titled: “An epigenetic change in a moth is generated by temperature and transmitted to many subsequent generations mediated by RNA” present interesting results regarding long lasting transgenerational epigenetic inheritance of antennae phenotype in the Mediterranean flour moth, Ephestia kuehniella (Lepidoptera: Pyralidae). Despite the experimental protocols applied are adequate, the way these results are presented must be improved, both the revision of the epigenetic processes and citing, redaction and English grammar must be improved before this ms can be accepted for publication.

I provided several suggestions to fix English and grammar issues in the attached PDF file. There are several phrases that are too colloquial or unformal. In other section referencing what is said is needed. Moreover, the order of the ideas presented or the supporting elements for the arguments developed is insufficient or confusing. This is a drawback for this work as the clarity of the message to be addressed is lost.

Below I provide examples of things that must be changed.

1- In several sections you illustrate with examples in mammals and nematodes. Nonetheless insect epigenetic research literature is quite abundant. I recommend you to try to decant your reasoning and proposal towards insect examples closer to your model organisms.

2- Line 96 you must describe briefly pre and post transcriptional epigenetic mechanisms.

3- Examples of colloquial writing: “molecular essence”, please change for: molecular mechanism

4- Paragraph order, you should use the sand clock structure for your introduction and discussion. Some general elements are presented too late in the explanation of important ideas. Please order.

5- In line 167 you suggest that “the epigenetic phenomenon is transmitted “on the father’s and mother’s side, perhaps even better o the father side”. However later in the discussion you reflect that “Even though this type of inheritance is possible along both the paternal and maternal lines, we focused only on the paternal line. The paternal inheritance is easier to study because the preparation of sperms is more simple than the non-fertilized eggs.” (line 364). Thus, seems that is not issue of the extent of epigenetic inheritance in parental lines, but a (legitimate) pragmatic decision. If that is the case you must change the phrase in line 167.

6- In discussion, please cite your figures and tables when referring to your own results.

7- Few pertaining references are suggested for discussion section.

8- Several paragraphs in discussion belong together.

9- Regarding “Acknowledgements”. Crude sarcasm, hateful messages and bulling must be eradicated from the science practice. If authors have any issue to solve among themselves or with other people, that is not business of this ms' reviewers even less its future audience. Please focus on good science writing instead. This paragraph must be replaced for true contributing thanks, or not included at all.

Same comments were sent to Editors.

Reviewer #3: PONE-D-22-29568R1

An epigenetic change in a moth is generated by temperature and transmitted to many subsequent generations mediated by RNA

Pavelka et al.

Pavelka et al. studied how temperature and different foods influence a short antenna mutation in the flour moth over multiple generations. They claim that the antennal mutation is an epigenetic effect, and that there is low % of switching over multiple generations and in response to environmental stress. The overall premise is very interesting and they have some compelling results. However, I have serious concerns on how the manuscript is written. I found the Introduction and the findings to be overstated.

The Discussion is way too long, and parts of the Discussion were tangential to the experiment. I also had issues with the experimental design and the interpretation of the results.

The terminology describing the mutant and wild type antennal phenotypes was very confusing. A figure could be helpful to help the reader track the mutations and terminology. The free online illustration program Biorender could be helpful with this issue.

I had concerns about the experimental design. In some cases, such as the egg injections, no control injections were done. In the revision, the authors should carefully describe the treatments (treatment and control), number of replicate individuals, and mortality rates from the exposure.

The Discussion was entirely too long and speculative. The authors should only focus on explaining their results, and avoid extending the discussion beyond what they can directly address with evidence.

Line 49. Describe the epigenetic phenomenon. Is it DNA methylation at a particular site? What is the phenotypic effect?

Line 53. What does “this epigenetic effect” refer to? Is it the presence/absence of a small RNA or a DNA methylation difference? I am not following the changes.

Line 47-61. What is the actual question/hypothesis tested by the study?

Line 68-69. The question is phrased very broadly. Instead, the summary should focus on how the study provide evidence for an explanation for evolutionary processes.

Introduction

Lines 71-108. There are multiple paragraphs embedded in this first paragraph. Carefully consider your main points, separate them into different paragraphs, and support the main points. The way that it is written the introductory paragraph is a jumble of ideas.

Line 75. The sentence “We can explain it with an example.” sounds too premature and general. What is “it” that is being explained? I recommend building up the explanation further ahead.

Line 76-80. Is this a real example? If so, it should cite a reference. The way that it is written sounds very general and idealized.

Line 76. I would avoid a gender specific pronoun. I would use the pronoun “it” instead.

Line 86. New paragraph.

Line 95. New paragraph. By the end of the first paragraph, the reader would ideally know what the study will focus on as a question.

Line 109. No evidence is provided as to why DNA-bound proteins should be important.

Lines 114-131. The provided evidence on epigenetics seem to be primarily in mammals. To what extent are the mechanisms shared between mammals and invertebrates?

Line 132. “Transgenerational” is misspelled.

Line 133. What does “molecular essence” mean? Mechanism?

Line 141. Comma is needed after “same”.

Line 143. Change “conversions” to “converts”.

Line 146. The topic sentence abruptly changes the narrative. I am unclear how this paragraph contributes to the overall argument.

Line 151. Consider what the main argument is for the paragraph. The last sentence does not really relate to the opening sentence.

Line 161. Be more specific in building the argument. What is the main point for the paragraph?

Line 162-163. I am confused as to whether the short antenna or long antenna is considered a mutation. In lines 142-143, it sounds like the long antenna is the mutation.

Line 167. This sentence is confusing.

Line 167-175. State the questions and hypotheses motivating the study.

Line 171. “Without epigenetic effect” sounds confusing. Maybe a word is missing.

Line 172. How was transmission to future generations determined? Provide enough of the approach at the end of the Introduction to guide the reader in the key questions motivating the study and the overall approach. How were the hypotheses tested?

Line 174. What was the “epigenetic effect”.

Methods

Line 200. What is a “paramutant” flour moth? How does one identify these moths?

Line 205. “With changed phenotype” makes it seem like it is the mutant.

Line 206. The nomenclature is completely confusing. The first generation saWT and the changed may be different phenotypes, but I can’t tell.

Line 212. Part of the vagueness of the narrative is that there is limited understanding of the epigenetic modification.

Line 213. How was the male spermatophore analyzed?

Line 215. “The line without epigenetic information” seems too simplistic because there is little evidence to demonstrate the mechanism at the genomic and transcriptomic level. There could be epigenetic information at other sites.

Line 244-247. What was the control treatment? I don’t think I see any. Without a control injection, the introduction of the Total RNA could be testing the injury/wounding from an injection rather than the RNA itself.

Line 247. One-tree?

Line 249. There was little justification for these treatments examining the “environmental effect of food”. More justification is needed in the Introduction.

Line 250-252. I don’t follow why milled wheat grains or plain flour would differ in their effect on antennal length. The narrative does not explain why the authors chose these factors.

Line 256. I don’t think I follow the study question. Why were they followed for 20 generations? What was the replication for the treatment effect?

Line 263. What does “additive to food” mean? How many different medium were used? What does “additive by RNA” mean? What does “group mean”?

Line 264. The question posed in the Introduction should align with the statistical test. I am not following what the hypotheses are.

Line 270. Explain what “reversed” and “non-reversed” clutches mean.

Results

Line 278. Cut this line.

Lines 280-291. The Methods section should be written in past tense. Be careful to not shift tenses within the same paragraph.

Lines 285-291. Clearly discuss the significant results. The narrative is not specific enough. Clearly describe how each main factor tested affected the likelihood of long-antenna offspring.

Lines 295-297. What about using likelihood ratio tests, which can be used in a generalized linear model framework?

Line 305. The numbers for the generation should be subscript.

Line 309-310. What is an extremely reversed clutch? How can this be defined more precisely?

Line 316. What are “five cultures”?

Discussion

Line 329. I couldn’t find any narrative how the authors were able to separate total RNA from small RNA. Something seems to be missing.

Line 330-331. Sentence is awkward.

Line 354-355. I don’t follow how geldanamycin activity confirms that the effect was due to small RNA>

Lines 355-362. This passage clarifies it more. This information should be put into the Methods.

Lines 363-369. This passage is written very casually, in incomplete paragraphs.

Lines 366-367. This statement is unsubstantiated.

Lines 371-383. These lines should be cut or paraphrased dramatically.

Lines 388-389. Cut the unrelated discussion such as plant responses to wounding.

Lines 406-410. Cut the paragraph on DNA methylation.

Lines 421- 434. Keep the discussion focused on the major findings of the study.

7. PLOS authors have the option to publish the peer review history of their article (what does this mean?). If published, this will include your full peer review and any attached files.

Reviewer #2: No

Reviewer #3: No

---

## [Author Response · Author response to Decision Letter 1]

24 Aug 2023

The manuscript was returned to make corrections to the text and to add some citations. We tried to comply with all the requirements, or to provide an explanation (e.g. the objection that control experiments are missing, even though they are clearly stated and are quite extensive - we have also emphasized repeatedly in the text which experiments are involved). In addition, we tried to make the discussion more clear.

I have attached a file to the answer to the manuscript, images, etc., but of course I can copy it into this form as well.

Below I provide examples of things that must be changed.

1- In several sections you illustrate with examples in mammals and nematodes. Nonetheless insect epigenetic research literature is quite abundant. I recommend you to try to decant your reasoning and proposal towards insect examples closer to your model organisms.

I have supplemented the literature on insects.

2- Line 96 you must describe briefly pre and post transcriptional epigenetic mechanisms.

I completed.

3- Examples of colloquial writing: “molecular essence”, please change for: molecular mechanism

Corrected.

4- Paragraph order, you should use the sand clock structure for your introduction and discussion. Some general elements are presented too late in the explanation of important ideas. Please order.

I tried to make the changes according to the instructions.

5- In line 167 you suggest that “the epigenetic phenomenon is transmitted “on the father’s and mother’s side, perhaps even better o the father side”. However later in the discussion you reflect that “Even though this type of inheritance is possible along both the paternal and maternal lines, we focused only on the paternal line. The paternal inheritance is easier to study because the preparation of sperms is more simple than the non-fertilized eggs.” (line 364). Thus, seems that is not issue of the extent of epigenetic inheritance in parental lines, but a (legitimate) pragmatic decision. If that is the case you must change the phrase in line 167.

I have corrected it in the places indicated to make it clearer and more understandable.

6- In discussion, please cite your figures and tables when referring to your own results.

Completed.

7- Few pertaining references are suggested for discussion section.

I have added the links.

8- Several paragraphs in discussion belong together.

Merged.

9- Regarding “Acknowledgements”. Crude sarcasm, hateful messages and bulling must be eradicated from the science practice. If authors have any issue to solve among themselves or with other people, that is not business of this ms' reviewers even less its future audience. Please focus on good science writing instead. This paragraph must be replaced for true contributing thanks, or not included at all.

I understand that it is not allowed to say and write a lot these days, I lived my youth under the dictatorship of the communist party, now it is a different ideology, but the methods are similar, and I know that there is no point in arguing, that is why I changed the “Acknowledgements”.

Same comments were sent to Editors.

Reviewer #3: PONE-D-22-29568R1

An epigenetic change in a moth is generated by temperature and transmitted to many subsequent generations mediated by RNA

Pavelka et al.

Pavelka et al. studied how temperature and different foods influence a short antenna mutation in the flour moth over multiple generations. They claim that the antennal mutation is an epigenetic effect, and that there is low % of switching over multiple generations and in response to environmental stress. The overall premise is very interesting and they have some compelling results. However, I have serious concerns on how the manuscript is written. I found the Introduction and the findings to be overstated.

The Discussion is way too long, and parts of the Discussion were tangential to the experiment. I also had issues with the experimental design and the interpretation of the results.

We tried to fix everything according to the comments, although the discussion remained long.

The terminology describing the mutant and wild type antennal phenotypes was very confusing. A figure could be helpful to help the reader track the mutations and terminology. The free online illustration program Biorender could be helpful with this issue.

I had concerns about the experimental design. In some cases, such as the egg injections, no control injections were done. In the revision, the authors should carefully describe the treatments (treatment and control), number of replicate individuals, and mortality rates from the exposure.

The mortality rate after injections was high, about 80%, but we haven't calculated it exactly, that's why it wasn't mentioned.

The Discussion was entirely too long and speculative. The authors should only focus on explaining their results, and avoid extending the discussion beyond what they can directly address with evidence.

We have corrected and supplemented the specific points that were described.

Line 49. Describe the epigenetic phenomenon. Is it DNA methylation at a particular site? What is the phenotypic effect?

Every epigenetic phenomenon is not DNA methylation. The explanation that this is the effect of RNA is in the next following sentence.

Line 53. What does “this epigenetic effect” refer to? Is it the presence/absence of a small RNA or a DNA methylation difference? I am not following the changes.

The epigenetic effect is written in the text before this sentence. But I added an explanation.

Line 47-61. What is the actual question/hypothesis tested by the study?

We are testing an interesting epigenetic effect with an unexpected length after 40 generations and what is its cause. So we will insert this sentence there, although in my opinion it follows from the text.

Line 68-69. The question is phrased very broadly. Instead, the summary should focus on how the study provide evidence for an explanation for evolutionary processes.

The study is not evolutionary in nature. The final question is only a general warning that the studied phenomenon could be an adaptation mechanism. But we've reworded that sentence to make it clearer.

Introduction

Lines 71-108. There are multiple paragraphs embedded in this first paragraph. Carefully consider your main points, separate them into different paragraphs, and support the main points. The way that it is written the introductory paragraph is a jumble of ideas.

We tried to rewrite it so it's not a jumble.

Line 75. The sentence “We can explain it with an example.” sounds too premature and general. What is “it” that is being explained? I recommend building up the explanation further ahead.

Deleted.

Line 76-80. Is this a real example? If so, it should cite a reference. The way that it is written sounds very general and idealized.

I tried to write it differently.

Line 76. I would avoid a gender specific pronoun. I would use the pronoun “it” instead.

Corrected.

Line 86. New paragraph.

OK

Line 95. New paragraph. By the end of the first paragraph, the reader would ideally know what the study will focus on as a question.

I tried to supplement it.

Line 109. No evidence is provided as to why DNA-bound proteins should be important.

The role of histones in epigenetics has been a matter of research and debate for about twenty years. Since it is necessary to briefly inform the reader as far as possible about all known epigenetic phenomena in order to include the considered possibilities, histones are also mentioned there. It is not the aim of the study to document these known facts from the literature in detail, it is not a review. We edit the text according to the proposed wording in the PDF file.

Lines 114-131. The provided evidence on epigenetics seem to be primarily in mammals. To what extent are the mechanisms shared between mammals and invertebrates?

This is a correct factual note. We have added examples known from insects. (When I wrote this work more than twenty years ago, it was not known in insects, and now I forgot to add it.)

Line 132. “Transgenerational” is misspelled.

Corrected.

Line 133. What does “molecular essence” mean? Mechanism?

Corrected.

Line 141. Comma is needed after “same”.

Corrected.

Line 143. Change “conversions” to “converts”.

Corrected.

Line 146. The topic sentence abruptly changes the narrative. I am unclear how this paragraph contributes to the overall argument.

It is explained here what the goal is in the presented study, but I have changed the position of the paragraph.

Line 151. Consider what the main argument is for the paragraph. The last sentence does not really relate to the opening sentence.

I tried to fix it so that the two sentences are related.

Line 161. Be more specific in building the argument. What is the main point for the paragraph?

The main point of the paragraph is that transgenerational epigenetic inheritance usually lasts only a few generations, and in the described case it is different, and that similar epigenetic phenomena in other cases are probably related to RNA. I added a short connecting sentence there.

Line 162-163. I am confused as to whether the short antenna or long antenna is considered a mutation. In lines 142-143, it sounds like the long antenna is the mutation.

I corrected the text to make it clearer. I replaced the long sentence with two short ones.

Line 167. This sentence is confusing.

I explained that insects have olfactory receptors on their antennae.

Line 167-175. State the questions and hypotheses motivating the study.

I tried to rewrite it.

Line 171. “Without epigenetic effect” sounds confusing. Maybe a word is missing.

I wrote it differently

Line 172. How was transmission to future generations determined? Provide enough of the approach at the end of the Introduction to guide the reader in the key questions motivating the study and the overall approach. How were the hypotheses tested?

Hypothesis testing is described in the previous lines. I tried to write it more clearly.

Line 174. What was the “epigenetic effect”.

Added: "reverting to wild type".

Methods

Line 200. What is a “paramutant” flour moth? How does one identify these moths?

I wanted to use a synonym so that I didn't have the same expression in the same sentence twice. OK, it's clearly incomprehensible that paramutant is a synonym for epigenetic effect, it's spelled differently now.

Line 205. “With changed phenotype” makes it seem like it is the mutant.

Line 206. The nomenclature is completely confusing. The first generation saWT and the changed may be different phenotypes, but I can’t tell.

I wrote both again and as primitively as possible.

Line 212. Part of the vagueness of the narrative is that there is limited understanding of the epigenetic modification.

I don't know how to respond to that. Something in the spermatophore was causing an epigenetic effect, so we went after what it should be. I don't know why this simple fact is vague and misunderstood.

Line 213. How was the male spermatophore analyzed?

I used a more appropriate term for this sentence.

Line 215. “The line without epigenetic information” seems too simplistic because there is little evidence to demonstrate the mechanism at the genomic and transcriptomic level. There could be epigenetic information at other sites.

I rewrote this sentence.

Line 244-247. What was the control treatment? I don’t think I see any. Without a control injection, the introduction of the Total RNA could be testing the injury/wounding from an injection rather than the RNA itself.

The inspection was carried out thoroughly!!!! I stated there that injections of buffer alone are controls.

Line 247. One-tree?

One-three - I missed a letter, I understand it was a puzzle.

Line 249. There was little justification for these treatments examining the “environmental effect of food”. More justification is needed in the Introduction.

I added to the introduction, the sentence that we tested the influence of other stress factors, not only temperature.

Line 250-252. I don’t follow why milled wheat grains or plain flour would differ in their effect on antennal length. The narrative does not explain why the authors chose these factors.

Unsuitable food is stress, and moreover, ontogenetic development is prolonged. We added a sentence to the introduction.

Line 256. I don’t think I follow the study question. Why were they followed for 20 generations? What was the replication for the treatment effect?

We followed twenty generations and counted all the images in the selected ones. We monitored the epigenetic effect up to 40 generations, when the experiment was terminated. The answer to the question of why we observed this is that no one has yet recorded such a long duration of an epigenetic phenomenon in a morphological feature in sexually reproducing creatures. There is only something similar on TE (now added in the text), when I did it, it was not even known. Or you can say that because science sometimes investigates new and interesting things. I didn't understand the second question.

Line 263. What does “additive to food” mean? How many different medium were used? What does “additive by RNA” mean? What does “group mean”?

The group means are the averages of the percentage of wild saWT offspring (out of the total number of offspring) of each moth’s pair. Groups are defined within each experimental intervention (see Methods). For example within type of medium (experimental intervention) we distinguished four groups (Flour, LiCl, NaCl, and Wheat grains) according to the medium the larvae were fed). 

Line 264. The question posed in the Introduction should align with the statistical test. I am not following what the hypotheses are.

Line 270. Explain what “reversed” and “non-reversed” clutches mean.

A clutch was classified as extremely reversed if the percentage of reversed individuals in the clutch was greater than 12.6% (the way how we calculated this threshold is described in the manuscript). 

Results

Line 278. Cut this line.

Deleted.

Lines 280-291. The Methods section should be written in past tense. Be careful to not shift tenses within the same paragraph.

We changed it. The text is in the past tense.

Lines 285-291. Clearly discuss the significant results. The narrative is not specific enough. Clearly describe how each main factor tested affected the likelihood of long-antenna offspring.

Dunn’s post-hoc tests (Table 2) showed several significant pairwise differences (nine out of the total of 28 pairwise comparisons). 

Lines 295-297. What about using likelihood ratio tests, which can be used in a generalized linear model framework?

The mentioned paragraph describes the results of post-hoc pairwise comparison and was moved to the previous paragraph where other pairwise comparison are described, which should be much clearer. We believe that ANOVA with pairwise comparison is routinely used for comparison of means and it is appropriate here. We agree, however, that the likelihood ratio test is also possible. We performed this test and the overall picture is not different from the ANOVA test, so we kept ANOVA results in the text. 

Line 305. The numbers for the generation should be subscript.

Changed.

Line 309-310. What is an extremely reversed clutch? How can this be defined more precisely?

See our answer above. 

Line 316. What are “five cultures”?

What the cultures are in this case is explained in the subsection "Animals and breeding", yes, five cultures were selected, where each individual was evaluated. That's why five. Not even three.

Discussion

Line 329. I couldn’t find any narrative how the authors were able to separate total RNA from small RNA. Something seems to be missing.

We assume it was a small RNA, it didn't come out that way from this sentence, it's fixed.

Line 330-331. Sentence is awkward.

The sentence has been rewritten, perhaps it is less awkward.

Line 354-355. I don’t follow how geldanamycin activity confirms that the effect was due to small RNA>

Maybe it was written too expressively, I fixed it. But why small RNA and the effect of geldanamycin are probably related is in the rest of the paragraph.

Lines 355-362. This passage clarifies it more. This information should be put into the Methods.

No, I don't think that the description of possible mechanisms and the probable connection with Argonaute proteins belongs to the methodology. In addition, the use of geldanamycin at the time was completely intuitive, we were simply trying to see if it would do something by affecting the most well-known heat shock protein. At that time, no one knew about Argonaute proteins. That's why writing in the methodology how it was calculated seems insincere to me.

Lines 363-369. This passage is written very casually, in incomplete paragraphs.

I rewrote it and dropped something.

Lines 366-367. This statement is unsubstantiated.

I expressed better that this is only a hypothesis and an assumption.

Lines 371-383. These lines should be cut or paraphrased dramatically.

I shortened it. But if I were to dramatically shorten the next text, the similarity with the current study would cease to be clear. I would just have to write that it was a similar thing and quote.

Lines 388-389. Cut the unrelated discussion such as plant responses to wounding.

Deleted.

Lines 406-410. Cut the paragraph on DNA methylation.

Deleted.

Lines 421- 434. Keep the discussion focused on the major findings of the study.

The objectives of the study are described in other parts of the text, I really do not understand why they should be declared in this paragraph. However, this paragraph is not very important and I could delete it completely.

---

## [Decision Letter · Decision Letter 2]

14 Sep 2023

An epigenetic change in a moth is generated by temperature and transmitted to many subsequent generations mediated by RNA

PONE-D-22-29568R2

Dear Dr. Pavelka,

We’re pleased to inform you that your manuscript has been judged scientifically suitable for publication and will be formally accepted for publication once it meets all outstanding technical requirements.

Kind regards,

J Joe Hull, Ph.D.

Academic Editor

PLOS ONE

Additional Editor Comments (optional):

Reviewers' comments:

Reviewer's Responses to Questions

**Comments to the Author**

1. If the authors have adequately addressed your comments raised in a previous round of review and you feel that this manuscript is now acceptable for publication, you may indicate that here to bypass the “Comments to the Author” section, enter your conflict of interest statement in the “Confidential to Editor” section, and submit your "Accept" recommendation.

Reviewer #2: All comments have been addressed

2. Is the manuscript technically sound, and do the data support the conclusions?

Reviewer #2: (No Response)

3. Has the statistical analysis been performed appropriately and rigorously? 

Reviewer #2: (No Response)

4. Have the authors made all data underlying the findings in their manuscript fully available?

Reviewer #2: Yes

5. Is the manuscript presented in an intelligible fashion and written in standard English?

Reviewer #2: Yes

6. Review Comments to the Author

Reviewer #2: No further comments. Authors included main suggestions and addressed problems from previous versions.

7. PLOS authors have the option to publish the peer review history of their article (what does this mean?). If published, this will include your full peer review and any attached files.

Reviewer #2: No

---

## [Editor Report · Acceptance letter]

12 Dec 2023

PONE-D-22-29568R2 

An epigenetic change in a moth is generated by temperature and transmitted to many subsequent generations mediated by RNA 

Dear Dr. Pavelka:

I'm pleased to inform you that your manuscript has been deemed suitable for publication in PLOS ONE. Congratulations! Your manuscript is now with our production department. 

Kind regards, 

on behalf of

Dr. J Joe Hull 

Academic Editor

PLOS ONE